# Prevention of Type 1 Diabetes in Children: A Worthy Challenge?

**DOI:** 10.3390/ijerph20115962

**Published:** 2023-05-26

**Authors:** Diletta Maria Francesca Ingrosso, Maria Teresa Quarta, Alessia Quarta, Francesco Chiarelli

**Affiliations:** Department of Pediatrics, University of Chieti, Via dei Vestini, 66100 Chieti, Italy

**Keywords:** type 1 diabetes, prevention, screening, autoimmunity

## Abstract

Nowadays, the development of new immuno-therapeutic drugs has made it possible to alter the course of many autoimmune diseases. Type 1 diabetes is a chronic disease with a progressive dependence on exogenous insulin administration. The ability to intercept individuals at high risk of developing type 1 diabetes is the first step toward the development of therapies that can delay the process of β-cell destruction, thus permitting a better glycemic control and reducing the incidence of ketoacidosis. The knowledge of the main pathogenetic mechanisms underlying the three stages of the disease may be helpful to identify the best immune therapeutic approach. In this review, we aim to give an overview of the most important clinical trials conducted during the primary, secondary and tertiary phases of prevention.

## 1. Introduction

Type 1 diabetes (T1D) is a chronic autoimmune disease with an insufficient insulin release due to a destruction of pancreatic β-cells, resulting in hyperglycemia. Islet autoantibodies that develop as the result of β-cells damage include autoantibodies against insulin (IAA), insulinoma-associated antigen-2 (IA-2), glutamic acid decarboxylase (GAD), zinc-transporter 8 (ZnT8), and islet cell antibodies (ICA) [1]. Over the last decades the incidence of T1D has dramatically increased, especially in toddlers and preschool children, imposing a huge economic burden on the global health system. In 2021, Ogle et al. estimated more than 108,300 new diagnoses of T1D among children and adolescents under 15 years of age [2]. The median age at seroconversion is before 3 years of age. Genetic predisposition and environmental factors are the main risk factors that influence T1D onset.

The first susceptibility genes to be identified were human leucocyte antigens (HLA), responsible for up to 50% of diabetes cases [3]. The susceptibility haplotypes most correlated to T1D are DR3.DQ2 and DR4.DQ8 haplotypes [4]. Genome-wide association studies (GWAS) detected more than 50 non-HLA loci associated with the risk of development of T1D [5]. HLA and non-HLA loci (PTPN22, CTLA4, IL2RA and INS) concur with the heritability of T1D, for approximately 80% of cases [6]. The insulin gene (INS) is the strongest genetic factor correlated to T1D, which may affect the immune reactivity to insulin [7]. A genetic risk score (GRS) that incorporated 30 T1D-associated SNPs from both HLA and non-HLA loci was developed in order to evaluate the risk of T1D progression. The GRS was shown to be a strong predictor tool of progression from single to multiple autoantibodies, and to the clinical development of T1D in high-risk individuals [8].

Several studies demonstrated the pivotal role of environmental factors that act as triggers at different time points in the course of the disease. Interestingly, Eisenbarth proposed a model of the natural history of T1D, representing the β-cell mass as a function of age. The first step is characterized by genetic predisposition, followed by an environmental trigger that determines the onset of immunological abnormalities, up to the development of overt diabetes [3]. Insel et al. recently proposed a staging classification based on the presence of autoantibodies, glycemia level and the manifestation of signs and symptoms of diabetes [9] (Figure 1). Depending on the number and the islet autoantibody type in circulation, it is possible to define a high or low probability of developing T1D. In particular, a high risk of progression rate is reported among those who have multiple islet autoantibodies. Moreover, the presence of IAA and IA-2 accelerates the timing of progression to diabetes from the time of seroconversion [10].

## 2. The Road to Prevention

The adoption of Eisenbarth and Insel’s pathogenetic staging of T1D has provided not only a standardized taxonomy [11], but also the possibility to identify subjects at risk of developing T1D, thus paving the way to its prevention. About 15,000 children and young adults who are first- or second-degree relatives of individuals with T1D are screened annually for islet autoantibodies through TrialNet [12]. Through the years, in addition to studies in family members, genetic testing of those without familial predisposition also proved to be a good tool, since only up to 15% of cases of newly diagnosed T1D have an identifiable family history [13,14]. A formidable accomplishment of this approach comes from Bavaria, Germany in the so-called “Fr1da” study, where Ziegler et al. screened 90,632 pre-school children aged between 2 and 5 years during primary care visits. Of these children, 280 (0.31%) had 2 or more islet autoantibodies, 62 developed clinical T1D, and 2 had mild or moderate diabetic ketoacidosis [15]. Before this, smaller studies had assessed screening of school-age children only [16,17,18,19].

Another relevant example of a large trial in the general population is the study “The Environmental Determinants of Diabetes in the Young” (TEDDY), a prospective cohort study that monitored 8667 genetically at-risk children at 3- to 6-month intervals from birth from 2004, for the development of islet autoantibodies and T1D [20]. The long-term goal of the TEDDY study was the identification of infectious agents, dietary factors, or other environmental agents, including psychosocial factors, that could trigger or protect from T1D in genetically susceptible individuals. As Bonifacio et al. reported [21], the risk of multiple seroconversion in these children declines exponentially with age, since the 5-year risk of developing multiple islet autoantibodies was 4.3% at 7.5 months of age and declined to 1.1% at a landmark age of 6.25 years [21]. Since genetic factors seem to influence the risk only in the first year of life, a practical aspect of this age-related seroconversion is the possibility to have an “efficient window” for multiple Ab diagnosis and T1D prevention: the highest sensitivity and positive predictive value was achieved by autoantibody screening at 2 years, and again at 5–7 years of age [21]. In another recent study, where 20,303 children with an increased T1D risk from Finland, Germany, and USA were screened and followed up to 18 years of age, the time from seroconversion to diabetes diagnosis was increased by 0.64 years for each 1-year increment of diagnosis age. Single screening at the age of 10 years was 90% sensitive, with a positive predictive value of 66% for clinical diabetes. Screening at two ages (10 years and 14 years) increased sensitivity to 93%, but lowered the positive predictive value to 55% [22]. These studies not only demonstrate that screening programs for T1D are possible, but they also may reduce the clinical burden of ketoacidosis (DKA) thanks to an earlier diagnosis. DKA is a life-threatening event which occurs in 30–60% of children newly diagnosed with T1D [23,24]; the incidence of DKA has dramatically increased during the COVID-19 pandemic [25]. In Fr1da, the prevalence of DKA was less than 5%, much lower than 20–30% in unscreened German children [26].

As in many other chronic and non-communicable diseases, prevention consists of three levels of intervention: primary prevention, aimed at avoiding development of autoimmunity against islet autoantigens; secondary prevention, interfering instead with β-cell destruction after the seroconversion; and finally, tertiary prevention, which is focused on preventing the clinical complications of T1D, with the main goal of delaying at least their onset, thus preserving residual β cell-function (Figure 1 and Figure 2).

## 3. Primary Prevention

It is evident that primary prevention has a strong reason for being, since it concerns the first period in life where the influence of genetic and environmental factors is the highest in the process of immunological tolerance. Primary prevention should start from pregnancy, since the influence of several factors on the risk of developing T1D has been observed in this phase. A prospective cohort study in Denmark analyzed the association between prenatal gluten exposure and offspring risk of T1D. High gluten intake during pregnancy seems to be correlated with an elevated risk of the offspring developing T1D. Overweight/obese women may be more sensitive to gluten intake than younger women or women of normal weight. The mechanism that could be responsible for this effect is not known, but it could include increased inflammation and a complex interaction between diet, immune development, microbiota and intestinal permeability. However, more evidence is needed to confirm these suggestive trends [27]. The correlation between maternal weight and the risk of developing T1D has been analyzed in several studies. A population-based study from Sweden, which included 1,263,358 children who were followed from birth, demonstrated how a high maternal BMI during the first trimester increases the incidence of T1D in the offspring of parents without diabetes. Thus, the prevention of excess weight and obesity in women of reproductive age may contribute to a decreased incidence of T1D [28]. In addition, maternal infection during pregnancy is considered to be a risk factor for developing T1D. A meta-analysis based on studies conducted in Finland and Sweden showed that maternal gestational infection is associated with 32% increased odds of T1DM or islet auto-immunity in offspring [29]. In particular, enterovirus infection during pregnancy increased the risk of T1D in offspring by up to 54%. This effect could be explained by the molecular mimicry between β-islet cell antigens and pathogens [30]. Prenatal protective factors for the development of autoimmunity include maternal vitamin D intake. The study “Diabetes Autoimmunity Study in the Young” (DAISY) evidenced the protective role of maternal intake of vitamin D via food during pregnancy on the risk of the onset of T1D in offspring. This protective effect may be due to the suppression role of vitamin D on the inflammatory cytokines in utero life [31].

All RCTs for the primary prevention of T1D have included early life dietary interventions and antigen-based therapy in newborns and toddlers, with the aim being to delay the loss of tolerance against pancreatic autoantigens. Interestingly, Vehik et al. [32] suggested that environmental exposure can trigger T1D in subjects who are less genetically susceptible, since high-risk HLA genotypes are becoming less frequent over time in some populations.

Breast milk has been found to contain various anti-inflammatory bioactive molecules such as immunoglobulins, adipokines, oligosaccharides, insulin, lactoferrin, lysozyme, cytokines, beneficial bacteria and vitamins, which can give infants a lifelong immunity against many diseases, including T1D. As hypoadiponectinemia is a risk factor for gestational diabetes mellitus (GDM), it has been suggested that adipokines in breast milk could have a potential role in increasing β-cell function in newborns [33]. Despite this, the association between breastfeeding and T1D remains controversial. In a meta-analysis of 43 studies, breastfeeding demonstrated a weak protective effect on T1D risk [34]; however, it should always be encouraged in children at risk, as for the general population.

Instead, based on the evidence in animal models that early exposure to cow’s milk proteins may increase the risk of β-cell autoimmunity [35], the TRIGR (“Trial to Reduce IDDM in Genetically at Risk”) was the first to be proposed as a dietary RCT in 5156 infants at genetic risk for type 1 diabetes, with a high retention and documented protocol adherence [36]. Unfortunately, the use of a hydrolyzed formula compared with a conventional formula did not reduce the incidence of β-cell autoimmunity after 7 years. Interestingly, a cumulative incidence of 9.9% by the age of 6 years for multiple seroconversion in the control group was estimated. TRIGR study results are in contrast to data from the TRIGR pilot study [37] performed in 230 Finnish children, which reported that weaning to an extensively hydrolyzed formula in infancy was associated with an almost 50% reduction in the cumulative incidence of β-cell autoimmunity by the age of 10 years in similar children.

Another dietary intervention trial is The Finnish Dietary Intervention Trial for the Prevention of Type 1 Diabetes (FINDIA), which assessed the impact of complete avoidance of bovine insulin in infants in Finland. It randomized high risk infants to cow’s milk formula, whey-based hydrolyzed formula, or whey-based formula free of bovine insulin during the first 6 months of life. A reduced rate of development of one islet autoantibody was seen in the bovine insulin-free group [38].

T1D and celiac disease (CD) are autoimmune diseases with a clinical and pathogenic overlap. In addition, an early introduction of cereals in infants had been associated with T1D [39]. In the open randomized controlled BABYDIET study [40], 150 newborns with genetic high risk were randomly assigned to a first gluten exposure at the age of 6 months (control group) or 12 months (late exposure group), and were followed at 3 monthly intervals until the age of 3 years. Again, delaying gluten exposure until the age of 12 months did not substantially reduce the risk for islet autoimmunity.

Implementation in the diet of omega-3 fatty acids, whose anti-inflammatory properties are well known, has been associated with a lower risk of T1D [41,42,43]. The Pilot Trial “Nutritional Intervention to Prevent Type 1 Diabetes” (NIP) was a double-blind placebo-controlled study of omega 3 fatty acid supplementation with docosahexaenoic acid (DHA). The aim of this study was to prevent islet cell autoimmunity in infants with an increased risk of developing T1D and who entered the study in the third trimester of pregnancy, or infants aged younger than 5 months with a first-degree family member with T1D. Thus far, no effect on autoimmunity has been noted [44].

Insulin has also been employed as an antigen-based therapy. Antigen-based therapy is based on the evidence that the administration of antigens can lead to an immunological tolerance [45,46,47]. The GPPAD-POInT Study is a randomized, placebo-controlled, double-blind phase IIb study that aimed to induce immune tolerance to β-cell autoantigens through regular exposure to oral insulin from 29 to 32 months. The study objective is to determine whether daily administration of oral insulin from 4 months to 7 months until the age of 3 years old to children with elevated genetic risk for T1D reduces the cumulative incidence of β-cell autoantibodies and diabetes in childhood. No data are available as yet [48]. Previously, oral insulin therapy at a modest daily dose of 7.5 mg did not delay progression to type 1 diabetes in first-degree relatives with insulin autoantibodies and ICA, except for a subgroup of relatives with high levels of insulin antibodies [49].

Within the environmental factors, viral infections have a predominant role in the pathogenesis of T1D in children with an elevated genetic risk. A viral infection could both cause β-cell cytolysis or indirectly trigger progressive diabetes autoimmunity [50]. Above all, viruses that have been evaluated to be potentially involved in T1D pathogenesis—the persistence of silent and chronic enterovirus infections, especially Coxsackie virus B1 (CVB)—have been strongly associated with the appearance of islet autoantibodies and an increased risk of T1D [51]. For this reason, vaccines can be one of the strategies on the road to primary prevention of T1D, such as an enteroviral serotypes vaccine [52]. Recent preclinical studies have outlined the concept that a CVB vaccine prevents viral infection and diabetes induction in mice [53]. In humans, an enterovirus vaccine may be effective for the primary prevention of T1D by blocking the triggering of autoimmunity and the onset of immune-mediated beta cell dysfunction [54]. Interestingly, an association between the Rotavirus vaccination and the incidence of T1D was noted [55]. In a cohort study of 1,474,535 infants in the United States from 2001 to 2017, a 33% reduction in the risk of T1D was reported with the completion of the Rotavirus vaccine series compared to the unvaccinated [55], suggesting a potential role of the Rotavirus vaccination in the prevention of the disease.

Gut microbiota has a primary role in the pathogenesis of T1D, affecting intestinal permeability, molecular mimicry, and modulating the innate and adaptive immune system. In the TEDDY study, modest alterations of microbial composition have been reported in patients with autoantibodies or T1D, particularly in the higher numbers of genes involved in fermentation pathways and the production of Short Chain Fatty Acids (SCFA). Butyrate, a SCFA product, is well known for its role in gut epithelial integrity maintenance and promoting anti-inflammatory responses [56,57]. In a mice model of T1D, the administration of *Lactobacillus johnsonii* delayed the development and the progression of the disease [58]. For all of these reasons, exploring the role of the microbiome may provide new highlights into developing safe strategies to modulate immune regulation in infants and children.

Finally, although most of these trials have failed in their primary outcomes, we may draw several lessons from them: primary prevention trials in first-degree relatives and beyond are feasible and may show a great adherence, offering the opportunity to have a second chance at prevention (secondary) if children develop β-cell autoantibodies [58].

## 4. Secondary Prevention

Secondary prevention includes several therapeutic strategies that are implemented during stage 1 and 2 of T1D when the autoimmune process has already been established. This is characterized by the presence of at least two auto-antibodies against β-pancreatic cells, the absence of dysglycemic states and clinical manifestations [1]. In this stage, the β-pancreatic cell population is still minimally affected, allowing an optimal glycemic control, although metabolic alterations can be evidenced after oral (OGTT) and intravenous glucose tolerance tests (IVGTT). Therefore, the main purpose of secondary prevention therapies is to limit, contain and stop the autoimmune process, prolonging the duration of stage 1 and 2 of T1D and preventing the onset of the clinical phase of diabetes [59]. Secondary prevention therapies may act on two fronts: the first is based on the induction of immunotolerance toward autoantigens, while the second consists of an immunomodulatory function in order to block the mechanisms and cells of autoimmunity [60,61]. Regarding the first, several clinical trials have been carried out, analyzing the effects of prolonged administration of low doses of specific self-antigens on the development of autoantibodies against β-pancreatic cells.

The “Diabetes Prevention Trial-Type 1” (DPT-1) clinical trial evaluated the effects of subcutaneous and oral insulin administration in patients with stage 2 T1D. Eligible patients aged between 1 and 45 years and with familiarity for T1D were screened by autoantibody assay, and then by IVGTT and OGTT. Patients with autoantibodies and positive glucose tolerance test were considered high-risk, while those with positive autoantibodies and a normal glucose tolerance test were considered medium-risk. The study population was then divided into a control group and two intervention groups, respectively, following subcutaneous insulin therapy at a dosage of 0.125 IU/Kg 2 vv/day combined with continuous intravenous insulin administration for 4 days at baseline and at T + 12 months (high-risk group), and one with oral insulin at a dosage of 7.5 mg/day (medium-risk group). Unfortunately, no results were obtained regarding the prevention of T1D in either intervention group compared with the placebo, although a subgroup with a high titer of IAA treated with oral insulin has shown a significant reduction in the incidence of T1D compared with the placebo [62,63,64]. The TrialNet study, conducted in a population with at least two autoantibody positivity and alterations at IVGTT, who were administered oral insulin at a dosage of 7.5 mg/day, showed no improvement in terms of prevention compared with the placebo; however, an insulin-treated subgroup highlighted a delay in the diagnosis of overt diabetes and a reduction in incidence compared with the placebo [65]. In the “Belgian Parenteral Study” trial, a test was carried out to discover whether regular administration of ultrarapid subcutaneous insulin before carbohydrate-rich meals may contribute to the maintenance of functional β-cell integrity in patients with autoantibody positivity and without alterations of glucose tolerance. A reduction in the incidence of T1D was observed compared with the placebo, albeit it was not statistically significant [66]. Intranasal insulin administration was then evaluated in subjects with familiarity for T1D and autoantibody positivity. The “Intranasal Insulin Trials” (INIT-1, INIT-II) did not demonstrate a clear benefit in the absence of adverse reactions [67,68]. The “Diabetes Prediction and Prevention Study” (DIPP) trial was performed in a population with a genetic predisposition for developing T1D and positivity for at least two autoantibodies through intranasal insulin administration; however, this trial was stopped due to a lack of benefit, even though the safety of intranasal insulin was demonstrated [69,70].

Another autoantigen tested in experimental trials has been GAD. In the clinical trial “Diabetes Prevention-Immune Tolerance” (DIAPREV-IT), the effect of administration of GAD conjugated to aluminum hydroxide (GAD-alum) was studied in normoglycemic children with anti-GAD antibodies and another type of positive antibody. This study showed differences that, although not statistically significant, compared with the placebo, since the treatment would result in an increase in Treg lymphocytes that would limit the action of anti-GAD65 T-lymphocytes; the safety of the drug was also demonstrated, but efficacy needs to be improved [71,72]. The main limitation of preventive therapies based on the immunotolerance mechanism is that the action of T-lymphocytes is not directed toward a single self-antigen, but is a complex reaction involving numerous self-antigens [73]. Another pharmacological approach is based on the immunomodulation through the administration of antibodies directed against specific targets. The main effectors of adaptive immunity are T and B lymphocytes; immunomodulatory therapies can act on these two cell lines by regulating their activity, which is typically unbalanced in autoimmune processes [74,75]. The effect of monoclonal antibodies against the TCR receptor complex of CD3+ T-lymphocytes was evaluated. The TrialNet study highlighted that Teplizumab, an anti-CD3 monoclonal antibody, delays the onset of clinical T1D in stage 2 patients with autoimmunity and dysglycemia [76]. The long-term effects of this treatment are the expansion of Treg and control of effector T-lymphocytes. It has been shown that Teplizumab is able to reduce the risk of disease progression to stage 3 by nearly 60% [77,78]. Teplizumab was approved by the Food and drug Administration (FDA) on November 2022 for adults and children of at least 8 years, and is the first approved drug that delays the onset of T1D. Therapy with Teplizumab consists of infusion of the drug intravenously once daily for 14 consecutive days. The main side effects are leukopenia, skin rash and headache [79].

Abatacept is a chimeric protein consisting of the extracellular Cytotoxic T-Lymphocyte Antigen 4 (CTLA4-Ig) domain bound to the Fc portion of human IgG1, which suppresses T-lymphocyte activity. Abatacept binds to signal molecules (CD80 and CD86), which, through binding to their respective receptors on T-lymphocytes, result in their activation and proliferation [80,81]. The Abatacept TrialNet study was conducted in a sample with autoimmunity and without dysglycemia, and is still ongoing. Immunomodulatory therapies that are directed toward B-lymphocytes act through limiting β-cell damage and regulating pancreatic T-lymphocyte infiltration [82,83]. Preclinical studies in mouse models have shown that depletion of B-lymphocytes by anti-CD20 antibodies (Rituximab) would correlate with the arrest of the autoimmune process. The major limiting factors are the important side effects related to B-cell depletion, such as increased susceptibility to opportunistic infections. The use of anti-CD20 antibodies also target the IL-10-producing subpopulation of regulatory B-cells (Breg), which play an important function in the mechanisms of immunotolerance. Finally, depletion of the B population also indirectly affects the Treg lymphocyte response. A possible strategy in order to limit these side effects could be the use of drugs that selectively target islet-responsive B-lymphocytes, and the use of combination therapies with other drugs that limit the side effects of anti-CD20 antibodies [84].

Regarding other types of pharmacological approaches, the effect of cyclosporine has been investigated, although limited data exist. It would seem that cyclosporine decreases the incidence of T1D in patients with autoimmunity and familiarity through its immunosuppressive effect on T-lymphocytes. However, considering the many associated side effects such as nephrotoxicity, its long-term use would be inappropriate [74]. In addition, nonpharmacological approaches such as the administration of high doses of vitamin B3, which would reduce free radical-induced damage to β-cells, have been evaluated in “The Deutsche Nicotinamide Intervention Study” (DENIS) and “The European Nicotinamide Diabetes Intervention Trial” (ENDIT) clinical trials. Although a positive effect was shown in animal models, in patients with familial and autoimmunity, no preventive benefit was shown in comparison with the placebo [85,86,87]. A possible preventive effect of vitamin D with omega-3 fatty acids was evaluated in the POSEIDON trial, evaluating their combined anti-inflammatory and immunomodulating effect [88].

## 5. Tertiary Prevention

At the time of diagnosis, more than 80% of β-cells mass is destroyed by specific CD4+ and CD8+ T-cells as well as B-cells; for this reason, a lifelong exogenous insulin is needed [89]. Tertiary prevention includes several strategies, with the main goals of preserving residual β-cells, thus reducing exogenous insulin requirement, and delaying the onset of complications. Several studies have demonstrated a correlation between lower HbA1c values and preserved C-peptide levels, with a reduction in long-term complications and better metabolic control [23].

In the last 12 years, various RCTs have been conducted in order to detect immune therapies that may ameliorate β-cells function in patients with T1D [90].

A single-blinded randomized placebo-controlled study demonstrated a preservation of C-peptide in patients with established T1D, through the combination of anti-thymocyte globulin (ATG) and pegylated G-CSF for 12 months following treatment. This study suggested an improvement of β-cells function up to 12 months following ATG and G-CSF treatment. This combination may affect the immune system, causing a reduction of CD4+ T cells and a preservation of CD8+ T cells [91]. The Type 1 Diabetes TrialNet Study Group performed a three-arm, randomized, double-masked, placebo-controlled trial in which the primary outpoint was to evaluate the area under the curve (AUC) of C-peptide by comparing ATG and ATG + G-CSF treatments. The group that received a low dose of ATG (2.5 mg/kg) had a better AUC C-peptide at 1 year than in those receiving ATG and G-CSF (respectively, 2.5 mg/kg and 6 mg subcutaneously every 2 weeks for 6 doses). At 1 year, HbA1c was lower in both treated groups, despite no significant differences in HbA1c levels, which were registered confronting ATG and ATG + G-CSF groups versus the placebo group [92].

The T1GER study was a phase 2 trial that tested Golimumab, a human monoclonal antibody specific for TNF alpha, among children and young adults with a recent T1D diagnosis. At 52 weeks, comparing golimumab versus the placebo group, a preservation of C-peptide and a reduction of the total daily insulin was documented. Despite no significative differences reported in Hb1Ac between the two groups, golimumab was correlated with a longer duration of partial remission characterized by minimal insulin production, which provides a glycemic control [93]. Beneficial effects were reported in the golimumab group after 2 years following the discontinuation of therapy. In particular, golimumab arrested β-cells loss, preserving their function while undergoing treatment [94]. The activation of lymphocytes T cells requires two stimulations: the first one is the interaction between the antigen, presented by the major histocompatibility complex (MHC) or the antigen-presenting cells (APC) and the T cells receptor. The second stimulation is a costimulatory signal between CD80/86 and CD28, which has a pivotal role in the full activation and survival of T cells. Abatacept prevents the costimulatory signal binding CD80/86, thereby sending an inhibitory signal inside the T cells. Some authors demonstrated that abatacept (CTLA4-Ig) preserves β-cells function in patients with a recent onset of T1D, slowing the decline of C-peptide during the 2 years of treatment by 9.6 months [95]. IL-21 cytokines stimulate CD8 + T lymphocytes, which play an important role in the pathogenesis of β-cells damage and the onset of T1D [96].

Moreover, the GLP-1 receptor agonist reduces β-cells stress and apoptosis, increases proinsulin/insulin ratio, and preserves insulin release glucose induced by stress factors. A double-blind randomized phase 2 trial studied the combination of anti-IL-21 and liraglutide in 308 patients with a recent T1D onset. This trial reported a better endogenous insulin secretion, assessed as a smaller decline in MMTT-stimulated C-peptide concentration, and a glucose metabolism improvement in the treated group from baseline to 54 weeks [97]. IL-2 plays an important role in the immune homeostasis, activating CD4 + FOXP3 + Tregs which limit the action of autoreactive anti-β-cell T effector cells (Teff). T1D pathogenesis susceptibility is due to a lacking regulation of Teffs by Tregs, which promote autoreactive Teffs expansion. Previous studies, such as DILT1D and DILfrequency performed in adults with T1D, provided reassuring information about the safety of aldesleukin administration [98,99]. The “Interleukin-2 Therapy of Autoimmunity in Diabetes (ITAD)” is an ongoing clinical phase 2, double-blind and placebo-controlled trial that included children and adolescents within 6 weeks from a T1D diagnosis. A total of 45 participants were included and randomized to receive an ultra-low dose of IL-2 (aldesleukin) or placebo for 6 months. The main goal of the ITAD trial is to assess the efficacy of aldesleukin treatment on β-cells function, as evaluated by C-peptide on a dried blood spot [100]. As previously stated, Teplizumab has been currently approved by FDA in children above 8 years of age as a disease-modifying therapy. Several trials have studied anti-CD3 monoclonal antibody efficacy in reducing insulin deterioration and β-cells damage [101,102]. Protégé was a phase 3 trial with 516 enrolled patients with a new onset of T1D who would receive Teplizumab administration. At 2 years, a reduced loss of AUC mean C-peptide was documented among those who received a 14-day full dose regimen of Teplizumab [103]. Moreover, the Autoimmunity-Blocking Antibody for Tolerance (AbATE) trial studied the action of Teplizumab in patients recently diagnosed with T1D for 7 years. At 1 year, a reduced loss of C-peptide was detected in drug treated responders, which was characterized by an augmentation of programmed cell death protein 1-positive central memory and CD8+ T cells. In addition, otelixizumab, another anti-CD 3 humanized antibody, has shown to affect the disease course only if administered in a high dose, despite stimulating an EBV reactivation. At the same time, a lower dose was not associated with a β-cells protection. The best effectiveness in preserving β-cells function is correlated with the positivity of IAA in patients with a recent T1D diagnosis [104,105]. It is anticipated that gene therapy will be a potential cure for T1D. The objective of gene therapy is based on transforming alternative cells in β-cells, thus preventing an attack on the immune system. Xiao et al. reprogrammed α-cells in β-cells through the infusion of an adeno-associated virus carrying Pdx1 and Maf, in autoimmune non-obese diabetic (NOD) mice. Pdx is a transcription factor required for β-cell maturation and proliferation, whereas MafA is a transcription factor that is able to regulate insulin expression [106]. The normal blood level in NOD mice lasts up to 4 months, but this result can be affected by the mouse lifespan. However, it is not possible to predict the immune system response in humans. Moreover, some studies focused on the identification of mechanisms able to stop the immune system from damaging β-cells. In the literature, the role of the antiapoptotic gene A20, which normally inhibits NF-κB activation was demonstrated. The overexpression of the A20 level by means of an adenoviral vector is able to improve islet transplantation outcomes, reducing the expression of inflammatory mediators. This effect is based on the inhibition of NF-κB and JNK/AP1 pathways [107,108]. Currently, the literature has described only pre-clinical trials on gene-therapy; although the results are very promising, more studies are needed.

## 6. Conclusions

T1D is a chronic autoimmune disease whose course is characterized by a pre-clinical and a clinical phase. The aim of prevention is to intervene during the early development of the disease by slowing or halting its progression. The first step in achieving valid results in prevention is to strengthen screening in the general population, and make them a part of the “Standard of Care”. Screening strategies could allow to identify individuals at risk for T1D in order to enroll them in clinical trials. Another possible benefit of screening programs is the purpose of reducing the incidence of DKA at the time of diagnosis [109], as well as the opportunity to offer psychological, emotional and social support to children and families at the time of diagnosis [24]. The importance of clinical trials is to identify new therapies that are safe and effective in changing the natural history of disease evolution. Although the alarming incidence of T1D in children increases the necessity of its prevention, there are several gaps and limitations concerning prevention strategies. Unfortunately, T1D is still considered a low prevalence disease in children, unlike other metabolic diseases such as obesity; the risk of false positives in screening tests is very high when the incidence of the disease is low [110]. For this reason, the lack of cost-effective and clinically advantageous screening programs makes it difficult to identify asymptomatic subjects and their participation in clinical trials. Moreover, the extraordinary technological advances in insulin therapy (including new insulin analogues, insulin pumps, continuous glucose monitoring systems, and automated insulin delivery systems) ensure that insulin still remains the “standard of care” in the treatment of diabetes, decreasing the rates of disease complications in proportion to glycemic control [111]. Besides insulin, only Teplizumab has been approved by FDA as the first drug that mildly delays the onset of T1D. Lastly, prevention strategies may lead to family and individual anxiety due to unwillingness to accept a treatment in the asymptomatic phases. We believe that this gap can only be conquered by a multidisciplinary approach, which provides not only good clinical practice but also psychological support to families.

A pivotal element in identifying possible new therapeutic strategies is to improve the knowledge of molecular mechanisms underlying the pathogenesis of T1D. A better understanding of these mechanisms could make it easier to identify possible targets for new immunological therapies. Although numerous clinical trials have not highlighted significant results, a winning strategy may be to use combination immunotherapies that act on several fronts. At present, several clinical trials have been conducted or are still ongoing; however, further studies are needed, and although the way to find safe and effective preventive therapies is still far, the obtained results are very promising.

## Figures and Tables

**Figure 1 ijerph-20-05962-f001:**
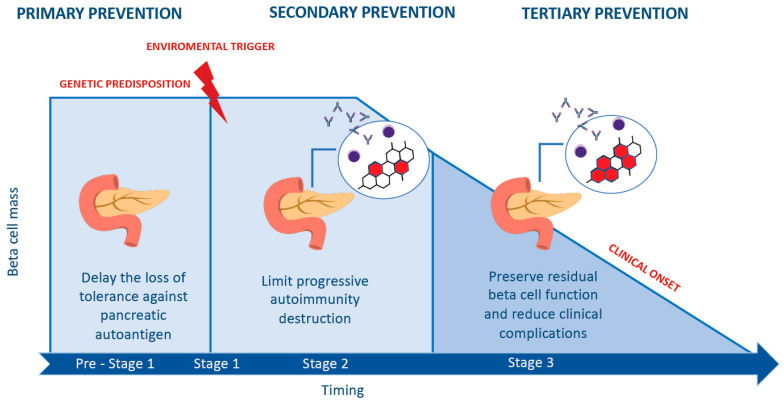
Types of prevention according to T1D stages.

**Figure 2 ijerph-20-05962-f002:**
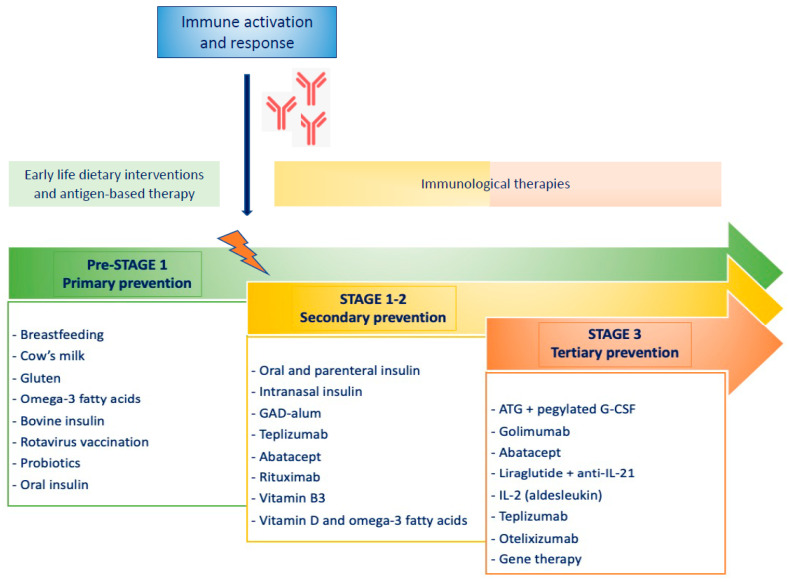
Main prevention and treatment options for different stages of T1D.

## Data Availability

All data are contained within the article.

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
