# Peer review of "Prevention of Type 1 Diabetes in Children: A Worthy Challenge?"

_ijerph, 2023, doi:10.3390/ijerph20115962_

Round 1
Reviewer 1 Report
Type of the Paper (Article
In the manuscript entitled " Prevention of Type 1 Diabetes: A Worthy Challenge?", the authors show overview of the most important clinical trials conducted during the pri-14 mary, secondary and tertiary phases of prevention. This experiment fits the aims and scope of the journal and needs some revision.
Please write the limitation in prevention strategy.
One important step in the pregnancy. Please discuss how we can manage and prevent diabetes during pregnancy
Discuss about the Prevention of T1D by gene therapy.
Discuss about mother milk hormones (novel finding) such as adiponectin and other factors.
Line 141: It randomized high risk infants to cow’s milk formula, whey-based hydrolyzed formula, or whey-based formula free of bovine insulin during the first 6 months of life. A reduced rate of development of one islet autoantibody was seen in the bovine insulin free group (39). Please transfer this part to Line 114(cow milk part).
Line 157: author mentioned “Above all viruses”, what type of virus?
Line 160: discuss vaccines and viruses. Please give an example in addition to Rotavirus. Please write the correlation between virus infections with diabetes.
Please write key findings in abstract.
Reviewer 2 Report
Ingrosso et. Al summarized the most important clinical trials conducted during the primary, secondary and tertiary phases of prevention.
Scientists working in this field are going to benefit form this review by having a well organized report of the available studies.
Clarity of the exposed topics is enhanced by ensuring consistent of the format of reported numbers, and by ensuring all the abbreviations are stated the first time they are mentioned.
Please ensure consistency in reporting numbers, for example, avoid reporting 10.000 in one instance and 10 000 in another instance.
The references looks appropriate.
Reviewer 3 Report
I think Francesca Ingrosso et al nicely summed up the different prevention states of Type 1 Diabetes and the state of several approaches in regards to treatment.
However, what I think will help this review would be an additional figure summarising the main findings or not established treamtent options for the different states of the diseases.
Round 2
Reviewer 1 Report
Thank you for corrections.